# FairImagen: Post-Processing for Bias Mitigation in Text-to-Image Models

**Zihao Fu**
The Chinese University of Hong Kong
zihaofu@cuhk.edu.hk

**Ryan Brown**
University of Oxford
blac0977@ox.ac.uk

**Shun Shao**
University of Cambridge
ss3047@cam.ac.uk

**Kai Rawal**
University of Oxford
kaivalyarawal45@gmail.com

**Eoin Delaney**
Trinity College Dublin
eoin.delaney@tcd.ie

**Chris Russell**
University of Oxford
chris.russell@oii.ox.ac.uk

## Abstract

Text-to-image diffusion models, such as Stable Diffusion, have demonstrated remarkable capabilities in generating high-quality and diverse images from natural language prompts. However, recent studies reveal that these models often replicate and amplify societal biases, particularly along demographic attributes like gender and race. In this paper, we introduce **FairImagen**[1], a post-hoc debiasing framework that operates on prompt embeddings to mitigate such biases without retraining or modifying the underlying diffusion model. Our method integrates Fair Principal Component Analysis to project CLIP-based input embeddings into a subspace that minimizes group-specific information while preserving semantic content. We further enhance debiasing effectiveness through empirical noise injection and propose a unified cross-demographic projection method that enables simultaneous debiasing across multiple demographic attributes. Extensive experiments across gender, race, and intersectional settings demonstrate that FairImagen significantly improves fairness with a moderate trade-off in image quality and prompt fidelity. Our framework outperforms existing post-hoc methods and offers a simple, scalable, and model-agnostic solution for equitable text-to-image generation.

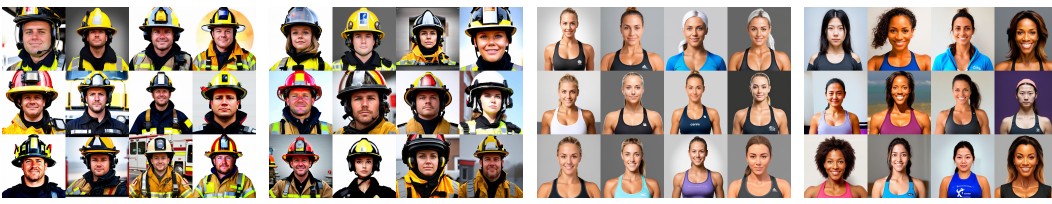

Firefighter (Base)  Gender Debias (FairImagen)  Yoga Instructor (Base)  Race Debias (FairImagen)

Figure 1: FairImagen mitigates demographic biases in text-to-image generation. Compared to baseline Stable Diffusion (Base), our method (FairImagen) produces more balanced representations across demographic attributes such as gender and race, while preserving visual quality and semantic fidelity.

## 1 Introduction

Recent advances in text-to-image generation have led to the widespread adoption of models such as Stable Diffusion [1, 2], DALL·E [3, 4], Imagen [5], and Parti [6]. These can produce photorealistic

---

[1] https://github.com/fuzihaofzh/FairImagen

39th Conference on Neural Information Processing Systems (NeurIPS 2025).

and diverse images from natural language prompts. These models leverage powerful vision-language encoders such as CLIP [7] to align textual and visual modalities, enabling open-ended image generation from arbitrary input. Due to their flexibility, scalability, and accessibility, these systems are increasingly integrated into applications across design, content creation, and interactive media [8, 9].

However, studies have shown that these generative models often replicate and even amplify social biases present in the training data [10–18]. For example, prompts such as: "a photo of a CEO" or "a nurse" typically yield images depicting white males and females, respectively, reflecting gender and racial stereotypes. These biases pose serious concerns regarding fairness, representation, and downstream harms, particularly as generative models are integrated into public-facing systems.

To mitigate such biases, researchers have proposed a variety of debiasing techniques. Methods fall into three main categories: **Prompt-based**, **Fine-tuning**, and **Post-hoc editing** (Table 1). Prompt-based approaches [10, 19, 20] modify the input to influence the model's output, but often require per image heuristic rewriting and manually-curated prompts. Fine-tuning methods [21, 13, 22] retrain or adapt parts of the model to encode fairness objectives, but they are computationally intensive and require access to model internals. Post-hoc editing methods [23, 24, 13] modify prompt embeddings at inference without updating model weights, offering a lightweight and deployment-friendly alternative. Each category exhibits differing trade-offs in terms of fidelity, interpretability, and generalizability.

We focus on **post-hoc editing** methods due to their simplicity and compatibility with a wide range of off-the-shelf diffusion models. Approaches such as SDID [23] and TBIE [24] demonstrated the feasibility of manipulating prompt embeddings to mitigate demographic bias. SDID identified a gender direction by subtracting CLIP embeddings of pairs of hand-crafted prompts; adding or subtracting this vector in generation. This heavily relies on the group bias being linearly separable and correctable via modifying embeddings in a single direction. This is inappropriate for debiasing involving more than two demographic groups, e.g., ethnicity. TBIE improves over SDID by applying PCA on CLIP embeddings of gender-related words to identify bias directions in a data-driven way. However, PCA is performed on simple gendered words without an explicit optimization criterion for content alignment. As a result, the debiasing process is often overly aggressive, removing not just demographic cues but also key semantic information. This results in semantic drift, loss of prompt fidelity, and unnatural image generation. Both methods offer limited control over the trade-off between fairness and fidelity, and neither generalizes well to intersectional prompts or unseen groups.

To overcome these limitations, we propose a novel post-hoc debiasing framework we call **FairImagen** (Fair Image Generation). It explicitly integrates Fair Principal Component Analysis (FairPCA) [25] into the Stable Diffusion pipeline and optimizes for semantic preservation while minimizing group-dependent variance. FairImagen operates in three stages: first, it extracts CLIP-based prompt embeddings; second, it applies a fairness-aware projection using FairPCA to remove group-dependent directions from both pooled and token-level embeddings; finally, it synthesizes images from the transformed embeddings using a modified Stable Diffusion decoder. To further enhance performance, we incorporate an empirical noise injection scheme to avoid overly neutralized outputs, and propose a unified cross-demographic debiasing formulation to jointly mitigate intersectional bias. Unlike existing post-hoc approaches, FairImagen offers precise control over the trade-off between fairness and content alignment. It is fully compatible with off-the-shelf diffusion models, supports multiple demographic attributes simultaneously, and preserves visual quality while effectively reducing bias.

Our contributions are as follows: (1) We introduce a post-hoc fairness framework that integrates FairPCA with diffusion-based text-to-image generation, enabling bias mitigation without model retraining. (2) We propose empirical noise injection to obscure residual demographic signals and improve fairness-performance trade-offs. (3) We develop a cross-demographic debiasing formulation that handles multiple protected attributes in a unified manor, avoiding over-pruning from sequential projections. (4) We conduct extensive quantitative and qualitative evaluations across gender, race, and joint debiasing tasks, demonstrating that our method outperforms existing post-hoc baselines.

## 2 Related Works

Existing debiasing methods for text-to-image generation can be categorized into three types: Prompt-based, Fine-tuning-based, and Training-free methods. As summarized in Table 1, no single category is universally superior; the choice of method often depends on the specific application scenario and deployment constraints.

| Criteria | Prompt-based | Fine-tuning | Post-hoc editing |
|---|---|---|---|
| Training-free | ✓ | ✗ | ✓ |
| Black-box compatible | ✓ | ✗ | ✓ |
| Low human effort | ✗ | ✓ | ✓ |
| Low computational cost | ✓ | ✗ | ✓ |
| Generalizable to new prompts | ✗ | ✓ | ✓ |
| Strong bias mitigation | ✗ | ✓ | ✓ |
| Preserves prompt fidelity | ✓ | ✓ | ✗ |
| Easy deployment | ✗ | ✗ | ✓ |

Table 1: Comparison of prompt-based, fine-tuning-based, and post-hoc editing methods for debiasing text-to-image generation.

**Prompt-based methods** mitigate bias by modifying the input prompts. Friedrich et al. [10] proposed Fair Diffusion using fairness-guided prompts constructed from demographic opposites. Sakurai and Sato [19] utilized LLMs to automatically detect and revise biased prompts. Bansal et al. [20] and Chuang et al. [26] examined ethical interventions and latent direction projection. Kim et al. [27] and Al Sahili et al. [28] developed learned fairness prompts, and Bianchi et al. [29] assessed the impact of biased prompts at scale. These methods are flexible but often rely on heuristic or external guidance for every single image. This can be somewhat opaque and laborious [20, 13].

**Fine-tuning based methods** update model parameters to enforce fairness. Li et al. [21] introduced Fair Mapping by training a linear projection layer. Zhang et al. [13] aligned prompt embeddings with fair visual examples. Shen et al. [22] applied a distributional alignment loss for fairness. Kim et al. [27], Orgad et al. [30], and Gandikota et al. [31] proposed fine-tuning specific modules or applying concept editing. Parihar et al. [32] incorporated interpretable latent directions and population-level optimization, respectively. These methods provide effective bias mitigation but often require costly model access and retraining.

**Post-hoc editing methods** avoid parameter updates and modify inference behavior. Zhang et al. [13] and Li et al. [23] manipulated prompt embeddings with CLIP-based or interpretable directions. Tanjim et al. [24] used PCA to subtract biased components. Friedrich et al. [10] employed classifier-free guidance alternations. Sadat et al. [33] explored sampling noise perturbation and conditioning annealing to reveal underrepresented concepts. Post-hoc filtering is also employed in some commercial systems [3]. These methods are deployment-friendly, take advantage of both prompt- and model-based strategies, and avoid extensive retraining or heavy prompt engineering.

## 3 Method

We propose a fairness-aware text-to-image generation framework, **FairImagen**. This framework integrates FairPCA [25] into Stable Diffusion [1, 2]. Our goal is to reduce social bias in image generation by modifying prompt embeddings prior to synthesis, while preserving semantic fidelity.

To estimate and remove demographic information from prompt embeddings, we begin with a small training set of natural language prompts, each annotated with protected attributes such as gender or race (e.g., "a lady playing computer," "an Asian man holding a phone"). These prompts are used to construct a FairPCA projection matrix that suppresses group-specific directions while retaining core semantic content. The learned projection is then applied at inference time to unseen prompts, making FairImagen entirely training-free with respect to the diffusion model.

The framework consists of three main components: (1) Prompt Embedding Extractor, (2) Fair Representation Transformer, and (3) Image Generator. In addition, we propose an empirical noise injection scheme to prevent overly neutralized outputs and a unified cross-demographic debiasing formulation to jointly mitigate intersectional bias.

### 3.1 Prompt Embedding Extraction

To construct the FairPCA projection, we begin with a small training set of natural language prompts $\mathcal{P} = \{p_1, \ldots, p_n\}$, each annotated with a protected attribute label $a_i \in \mathcal{A}$. These prompts are designed to be demographically informative yet semantically neutral (e.g., "a lady playing computer," "a Black man riding a bike"), and serve as the foundation for identifying group-dependent components in the embedding space.

Given a prompt $p \in \mathcal{P}$, we first encode it using a pre-trained CLIP model [7]. Let $\{w_1, \ldots, w_T\}$ be the tokenized prompt, where $T$ is the number of tokens. The encoder outputs a token-level embedding matrix $E_p \in \mathbb{R}^{T \times D}$, where $D$ is the embedding dimension, and a pooled embedding $\bar{E}_p \in \mathbb{R}^D$. The pooled embedding is computed as the mean of the token embeddings: $\bar{E}_p = \frac{1}{T} \sum_{t=1}^{T} E_p[t]$. These representations are extracted from the Stable Diffusion text encoder. Let $\mathcal{P} = \{p_1, \ldots, p_n\}$ denote a set of prompts, each associated with protected attribute labels $a_i \in \mathcal{A}$. For each attribute $a$, we organize the pooled embeddings by group:

$$X = \{\bar{E}_{p_i}\}_{i=1}^n \in \mathbb{R}^{n \times D}, \quad Z = \{z_i\}_{i=1}^n \in \{0, 1\}^{n \times G},$$

where $z_i$ is a one-hot group indicator for the attribute $a_i$, and $G = |\mathcal{A}|$. These grouped embeddings are used to estimate the bias direction and define fairness-aware projections.

## 3.2 Fair Representation Transformer

We use Principal Component Analysis (PCA) to approximate the original prompt embedding space with a lower-dimensional subspace that preserves semantic information. Specifically, we seek a projection matrix that can faithfully reconstruct the original embeddings from a reduced set of basis directions.

Let $P \in \mathbb{R}^{D \times d}$ be a projection matrix, where $d < D$. Classical PCA solves the following reconstruction objective:

$$\underset{P \in \mathbb{R}^{D \times d} : P^\top P = I}{\arg\min} \sum_{i=1}^{n} \left\| \mathbf{x}_i - PP^\top \mathbf{x}_i \right\|_2^2 \equiv \underset{P \in \mathbb{R}^{D \times d} : P^\top P = I}{\arg\max} \operatorname{Tr}(P^\top XX^\top P), \tag{1}$$

where $X = [\mathbf{x}_1, \ldots, \mathbf{x}_n]^\top \in \mathbb{R}^{n \times D}$ is the matrix of pooled prompt embeddings. The left-hand side minimizes the total squared reconstruction error of projecting the data onto a $d$-dimensional subspace, while the right-hand side expresses the equivalent trace maximization formulation.

To further ensure fairness, we also require that the projection removes demographic signals by aligning with the null space of group-dependent variation. Specifically, we define the group-dependent feature matrix $B = Z^\top X \in \mathbb{R}^{G \times D}$, where $Z \in \{0, 1\}^{n \times G}$ is the group indicator matrix. Each row of $B$ captures the mean embedding direction for a specific demographic group. By constraining the projection matrix $P$ to lie in the null space $\mathcal{N}(B)$, we ensure that the resulting representations are orthogonal to any direction that separates groups in the embedding space, thereby eliminating linear demographic signals.

Building on this intuition, FairPCA [25] incorporates a fairness regularization term into the PCA objective, yielding the following formulation:

$$\min_{P^\top P = I} - \operatorname{Tr}(P^\top \Sigma_X P) + \lambda \|BP\|_F^2, \tag{2}$$

where $\lambda$ is a hyperparameter controlling the trade-off between reconstruction quality and fairness. The first term ensures that the projection subspace remains a good approximation of the original feature space. The second term penalizes the degree to which the projected embeddings retain group-specific components, thereby reducing demographic separability. Once the projection matrix is obtained, we apply it during inference to both the pooled and token-level prompt embeddings:

$$\bar{E}_p' = PP^\top \bar{E}_p, \quad E_p' = E_p PP^\top.$$

## 3.3 Empirical Noise Injection

To further enhance diversity and realism, we introduce an empirical noise injection mechanism that perturbs representations along estimated group-dependent directions. It prevents the generated output from becoming overly neutral (e.g., generating a man who appears feminine; see further discussion in Appendix E). Let $\mathcal{G}$ denote the set of protected groups (e.g., $\mathcal{G} = \{\text{Male}, \text{Female}\}$), and let $g \in \mathcal{G}$ be a particular group. For each group $g$, we compute its empirical bias direction as

$$v_g = \frac{1}{|X^{(g)}|} \sum_{\bar{E}_p \in X^{(g)}} \bar{E}_p - \bar{E},$$

where $X^{(g)}$ is the set of pooled embeddings belonging to group $g$, and $\bar{E}$ is the overall mean embedding across all groups. We define an empirical distribution $\mathcal{D}_g$ as the set of scalar projections of group-specific embeddings onto the bias direction:

$$\mathcal{D}_g = \left\{ v_g^\top \bar{E}_p : \bar{E}_p \in X^{(g)} \right\}.$$

Each value $\delta \in \mathcal{D}_g$ represents the magnitude of projection of an embedding onto the bias direction $v_g$, quantifying how strongly that embedding aligns with group-specific attributes. To inject noise in the inference stage, we sample $\delta \sim \mathcal{D}_g$ and apply the perturbation:

$$\bar{E}_p'' = \bar{E}_p' + \epsilon \cdot \delta \cdot v_g,$$

where $\epsilon$ is a tunable noise scale parameter. This procedure introduces controlled variability along biased directions to obscure protected group information while preserving semantic structure. We also conduct experiments with additional noise injection strategies, including mean-based, Gaussian, fixed-directional, and deterministic shift variants. A detailed comparison of these methods and their impact on fairness and image quality is provided in Appendix D. Among these strategies, empirical noise demonstrates the best overall performance.

### 3.4 Cross-Demographics Debiasing

The FairPCA framework [25] debiases multiple demographics by jointly encoding them into a multi-dimensional attribute matrix. It minimizes group-specific information across all attributes simultaneously via a single projection derived from a stacked group indicator matrix. However, when applied to image generation, this approach fails to adequately represent all demographics, as it forces features to be orthogonal to each group direction. Consequently, the model tends to preserve information aligned with only one group at a time, resulting in degraded contextual fidelity and the loss of important visual details in the generated images.

To overcome this, we propose a unified cross-demographic debiasing method that constructs a single attribute space based on the Cartesian product of all group combinations. For example, if the gender attribute has two groups {Male, Female} and the race attribute has three groups {White, Asian, Black}, we define a joint attribute space with six composite groups: $\mathcal{A}_{\text{joint}} = $ {White Male, White Female, Asian Male, Asian Female, Black Male, Black Female}. We then apply Fair Representation Transformer once over this joint attribute space. Therefore, our cross-demographic debiasing approach can debias all demographics simultaneously. We have also conducted experiments with alternative strategies for handling multiple demographic attributes, including stacking and sequential projection. For a comprehensive comparison of these cross-demographic debiasing methods, please refer to Appendix G.

### 3.5 Image Generator

After debiasing, we pass the transformed embeddings into a customized Stable Diffusion pipeline [1, 2], which supports external prompt embeddings. Specifically, we generate the image as: $I_p = \mathcal{G}(\bar{E}_p'', E_p')$, where $\mathcal{G}(\cdot)$ denotes the generation function, and $\bar{E}_p''$ and $E_p'$ are the pooled and token-level debiased embeddings.

## 4 Experiments

### 4.1 Dataset

The Winobias dataset consists of 46 professions, collected from the US Bureau of Labor Statistics, that are stereotypically considered "male biased" or "female biased" [31, 30, 34, 35]. In our experiments, we extend this list to 120 professions using publicly available lists[2]. Our list covers professions that have been found to be biased towards men (e.g., Janitor or CEO) and women (e.g., Nurse or Librarian) in generative AI systems [36].

---

[2] The full list is included in our supplementary material with the code. We manually extended the winobias list using a publicly available list of occupations from Wikipedia: `https://en.wikipedia.org/wiki/Lists_of_occupations`

## 4.2 Experimental Settings

Our modified pipeline extends HuggingFace's StableDiffusion3Pipeline to accept external embeddings and apply FairImgen at inference time. We generate images using classifier-free guidance with scale $w = 7.0$ and $T = 28$ diffusion steps. Images are generated in batches (12 per prompt), stitched, and evaluated with fairness and perceptual quality metrics. We split the dataset into a development set of 20 samples and use the remaining 100 samples as the test set. We tune all models on the development set to maximize the average (AVG) score and report their performance on the test set. We run all the models on a NVIDIA A100 GPU with 80 GB memory.

## 4.3 Evaluation

We report four scalar metrics: Fairness, Accuracy, MUSIQ, and their average.

Fairness is monitored using a lightweight pretrained facial attribute classifier from DeepFace [37]. This classifier detects and counts members of each group in every generated image. Next the distribution of counts is scored with

$$1 - \frac{\sum_i |p_i - \frac{1}{k}|}{2(1 - \frac{1}{k})}$$

following the normalized-deviation formulation of Teo et al. [38]. Here $p$ is a vector of group proportions and $k$ is the number of groups. A score of 1 indicates that all groups are generated at the same rate, while a score of 0 indicates that only one group is generated (e.g., all men when considering gender as the protected attribute and prompted to generate images of a CEO). We measure gender-based groups of male and female individuals alongside ethnicity-based groups of asian, black, latino hispanic, middle eastern, and white individuals. We also consider intersectional groups of gender and ethnicity throughout the paper.

Accuracy is measured using CLIPScore [39], which quantifies how closely a generated image matches its text prompt. Specifically, we compute the cosine similarity between the prompt embedding and the image embedding produced by CLIP (ViT-B/16 backbone). Following best practices reported by Hessel el al. [39], we multiply the CLIPScore values by 2.5 for scaling as the original scores typically fall between 0 and 0.4.

To assess the visual quality of an image, we use MUSIQ [40], which is a no-reference perceptual-quality model trained on millions of aesthetic ratings. This metric was selected due to its flexibility, as it can work with native resolution images with varying sizes and aspect ratios.

## 4.4 Comparison Models

**Base** is the vanilla Stable Diffusion model, which directly generates images from the prompt without any fairness intervention.

**FairPrompt** follows [10, 19] by using human-designed prompts for each image. We evenly apply different prompts corresponding to protected groups for each individual image. This serves as an upper-bound performance baseline, as each prompt is specifically tailored for fairness.

**ForcePrompt** explicitly includes fairness-related instructions (e.g. "Please avoid gender bias.") in the prompt, directing the Stable Diffusion model to generate fair representations.

**SAL** [41, 42] uses Singular Value Decomposition to project the input representations into directions with reduced covariance with the biases.

**CDA** (Counterfactual Data Augmentation) [43, 44] replaces gendered words with their counterfactual counterparts, such as replacing "man" with "woman." We follow the CDA methodology to construct counterfactual samples and augment the dataset.

**TBIE** (Text-Based Image Editing) [24] applies PCA to gender-related words and performs debiasing along the identified principal components.

**SDID** (Self-Discovering Interpretable Diffusion) [23] computes a gender vector using the difference between gender-specific and gender-neutral embeddings, and injects this vector into the prompt embedding.

| | Gender Fairness | Accuracy | MUSIQ | Avg | | Race Fairness | Accuracy | MUSIQ | Avg | | Gender Fairness | Race Fairness | Accuracy | MUSIQ | Avg |
|---|---|---|---|---|---|---|---|---|---|---|---|---|---|---|---|
| Base | 0.167 | 0.785 | 0.574 | 0.509 | Base | 0.193 | 0.785 | 0.574 | 0.517 | Base | 0.163 | 0.193 | 0.785 | 0.574 | 0.508 |
| FairPrompt | 0.732 | 0.766 | 0.586 | 0.695 | FairPrompt | 0.444 | 0.752 | 0.566 | 0.587 | FairPrompt | 0.69 | 0.478 | 0.747 | 0.574 | 0.671 |
| ForcePrompt | 0.292 | 0.755 | 0.601 | 0.549 | ForcePrompt | 0.266 | 0.761 | 0.574 | 0.534 | ForcePrompt | 0.287 | 0.304 | 0.764 | 0.591 | 0.547 |
| SAL | 0.217 | 0.779 | 0.602 | 0.533 | SAL | 0.262 | 0.788 | 0.607 | 0.552 | SAL | 0.182 | 0.214 | 0.776 | 0.599 | 0.519 |
| CDA | 0.547 | 0.772 | 0.549 | 0.623 | CDA | 0.358 | 0.772 | 0.537 | 0.556 | CDA | 0.362 | 0.27 | 0.779 | 0.557 | 0.566 |
| TBIE | 0.35 | 0.782 | 0.567 | 0.566 | TBIE | 0.366 | 0.762 | 0.532 | 0.553 | TBIE | 0.40 | 0.286 | 0.776 | 0.546 | 0.574 |
| SDID | 0.507 | 0.776 | 0.553 | 0.612 | SDID | 0.37 | 0.77 | 0.537 | 0.559 | SDID | 0.223 | 0.256 | 0.782 | 0.556 | 0.52 |
| SDID-AVG | 0.315 | 0.783 | 0.562 | 0.553 | SDID-AVG | 0.361 | 0.769 | 0.544 | 0.558 | SDID-AVG | 0.352 | 0.28 | 0.778 | 0.553 | 0.561 |
| ITI | 0.27 | 0.769 | 0.528 | 0.522 | ITI | 0.214 | 0.77 | 0.53 | 0.504 | ITI | 0.32 | 0.235 | 0.747 | 0.467 | 0.511 |
| FairQueue | 0.197 | 0.809 | 0.621 | 0.542 | FairQueue | 0.118 | 0.736 | 0.631 | 0.495 | FairQueue | 0.0567 | 0.34 | 0.773 | 0.606 | 0.478 |
| FairImagen | 0.56 | 0.771 | 0.541 | 0.624 | FairImagen | 0.389 | 0.76 | 0.536 | 0.562 | FairImagen | 0.537 | 0.32 | 0.753 | 0.544 | 0.611 |
| FairImagen-T5 | 0.572 | 0.768 | 0.533 | 0.624 | FairImagen-T5 | 0.386 | 0.76 | 0.537 | 0.561 | FairImagen-T5 | 0.48 | 0.31 | 0.766 | 0.532 | 0.593 |
| FairImagen-OC | 0.573 | 0.767 | 0.534 | 0.625 | FairImagen-OC | 0.387 | 0.76 | 0.536 | 0.561 | FairImagen-OC | 0.482 | 0.311 | 0.766 | 0.532 | 0.593 |

| (a) Gender Debias. | (b) Race Debias. | (c) Gender + Race Debias. |
|---|---|---|

Table 2: Quantitative evaluation results for debiasing text-to-image generation across three settings: (a) gender, (b) race, and (c) both gender and race. FairImagen achieves the best overall performance among post-hoc methods, striking a strong balance between fairness, accuracy, and perceptual quality.

**SDID-AVG** extends the SDID [23] model by computing neutral embeddings through averaging the embeddings within each protected group.

**ITI-GEN** [13] extracts gender-related CLIP embeddings from images and adds them to the prompt embeddings prior to image generation.

**FairQueue** [45] rethinks prompt learning approaches by identifying abnormalities in early denoising steps. It proposes Prompt Queuing (using base prompts without sensitive attribute tokens in initial steps) and Attention Amplification (enhancing attribute representation in later steps) to modify cross-attention maps during generation, achieving competitive fairness while improving image quality.

**FairImagen-T5/OC**: To evaluate if our method is compatible with encoders beyond CLIP, we evaluate using alternative text encoders. **FairImagen-T5** replaces the CLIP text encoder with T5 [46], while **FairImagen-OC** uses OpenCLIP [47] instead of the original CLIP encoder. These variants demonstrate the generalizability of our fairness framework across different text encoding architectures.

### 4.5 Experimental Results

**4.5.1 Main Experiments.** We apply our model, as well as other baseline models, to generation tasks involving debiasing with respect to gender (§4.5.1), race (§4.5.1), and both gender and race simultaneously (§4.5.1). The results show that: (1) our proposed FairImagen model outperforms all postprocessing baseline models in terms of fairness scores across all three scenarios, demonstrating its effectiveness in mitigating bias in various contexts; (2) our proposed FairImagen model also outperforms all postprocessing models in terms of average (AVG) scores, indicating that it achieves the best balance among fairness, accuracy, and image quality; and (3) our model consistently outperforms all postprocessing baselines when debiasing both gender and race simultaneously, highlighting its strong capability in addressing multi-attribute bias mitigation. (4) Our proposed FairImagen model slightly lags behind other models in terms of Accuracy and MUSIQ. However, given the substantial improvement in Fairness, this trade-off is justified and considered worthwhile. (5) FairPrompt achieves the best performance across all experiments. It should be noted, however, that this model relies on manually designed prompts tailored to each individual image, which is both time-consuming and labor-intensive. As such, it serves primarily as an upper bound to illustrate the best possible performance a model can achieve on this task.

**4.5.2 Effect of Hidden Dimension on FairImagen Performance.** To investigate the impact of dimensionality reduction on the effectiveness of FairImagen, we vary the number of retained principal components (i.e., hidden dimensions) from 200 to 2000. Figure 2 shows performance trends across three scenarios: (a) gender debiasing, (b) race debiasing, and (c) joint gender and race debiasing. The results demonstrate a clear trade-off between fairness and other metrics as dimensionality varies. Notably, reducing the number of components tends to improve fairness scores, particularly in gender and race separately, but at the cost of reduced Accuracy and MUSIQ. In contrast, larger hidden dimensions preserve visual and semantic fidelity better, but may reintroduce bias. The joint debiasing setting (Figure 2c) further reveals the challenge of balancing fairness across multiple attributes simultaneously, with Fairness metrics for gender and race sometimes diverging. Overall, these

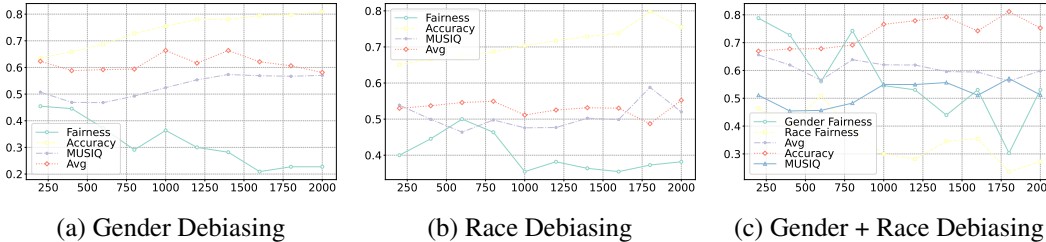

|                     |                     |                           |
| :-----------------: | :-----------------: | :-----------------------: |
| (a) Gender Debiasing | (b) Race Debiasing | (c) Gender + Race Debiasing |

Figure 2: Effect of hidden dimension size on FairImagen performance across different debiasing settings: (a) gender debiasing, (b) race debiasing, and (c) joint gender and race debiasing. Reducing the number of retained dimensions improves fairness but may reduce Accuracy and MUSIQ, highlighting the trade-off between fairness and semantic or visual fidelity.

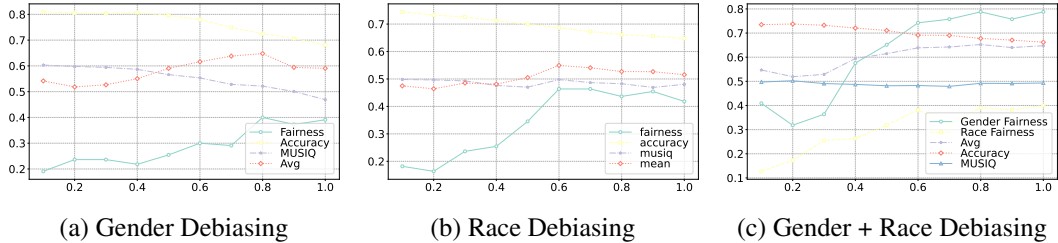

|                     |                     |                           |
| :-----------------: | :-----------------: | :-----------------------: |
| (a) Gender Debiasing | (b) Race Debiasing | (c) Gender + Race Debiasing |

Figure 3: Effect of e-noise on FairImagen performance across different debiasing settings: (a) gender debiasing, (b) race debiasing, and (c) joint gender and race debiasing. Increasing e-noise enhances fairness, especially in the joint setting, but introduces a moderate drop in Accuracy and MUSIQ. This highlights the need to balance fairness improvements with fidelity preservation when tuning noise.

results indicate that non-linear biases are more likely to occur in higher-dimensional spaces, and points towards the importance of selecting an appropriate dimensionality to achieve a desirable fairness-fidelity trade-off.

**4.5.3 Effect of Empirical Noise on FairImagen Debiasing.** We further examine the influence of the empirical noise parameter (e-noise) in the FairImagen framework, which controls the magnitude of perturbation added to simulate empirical distributions aligned with different protected attributes. As e-noise increases, the sampled directions more closely follow gender- or race-specific variations, enabling stronger debiasing effects. Figure 3 presents the performance across gender, race, and joint gender+race debiasing tasks under varying noise levels from 0.0 to 1.0. Higher e-noise values significantly improve fairness metrics, particularly in the joint debiasing setting, where both gender and race fairness steadily increase. However, this comes with a modest trade-off in Accuracy and MUSIQ, suggesting a balance must be struck between fairness gains and fidelity preservation. These results validate the effectiveness of controlled empirical noise injection in sampling representative latent directions that better approximate the real distribution of protected attributes.

**4.5.4 Qualitative Case Study.** To qualitatively assess our debiasing strategies, we examine the prompt "Generate a photo of a face of a CEO" under four FairImagen settings: Base, FairImagen-Gender, FairImagen-Race, and FairImagen-Gender+Race. As shown in Figure 4, the Base model predominantly generates white male faces, reflecting societal bias. FairImagen-Gender (Figure 4b) increases female representation while maintaining visual coherence. FairImagen-Race (Figure 4c) introduces more racial diversity, including Black and Asian CEOs. Combining both dimensions (Figure 4d) yields broader diversity but also introduces visual inconsistencies in background and style, due to stronger empirical noise. This also highlights the trade-off: enhancing fairness can compromise visual stability and prompt fidelity.

**4.5.5 Evaluation on Occupations with Man/White Dominance.** To evaluate debiasing performance, we analyze occupation prompts that exhibit strong male and white biases in the baseline model. Figure 5 shows that the Base model produces predominantly male outputs, while FairImagen and FairPrompt significantly improve gender balance. Similarly, Figure 6 reveals strong white dominance

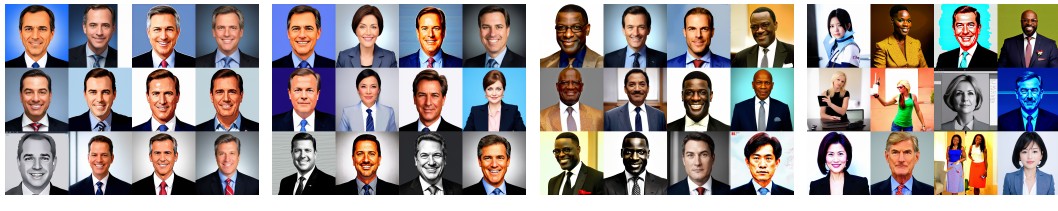

| (a) Base | (b) FairImagen-Gender | (c) FairImagen-Race | (d) FairImagen-G+R |

Figure 4: Generated results for the prompt "Generate a photo of a face of a CEO" under four FairImagen settings: (a) Base (no debiasing), (b) FairImagen-Gender, (c) FairImagen-Race, and (d) FairImagen-Gender+Race. Debiasing increases demographic diversity across gender and race dimensions. However, stronger debiasing—especially under intersectional settings—can introduce variation in background and style, reflecting a trade-off between fairness and visual consistency.

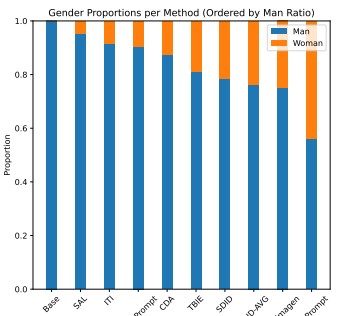 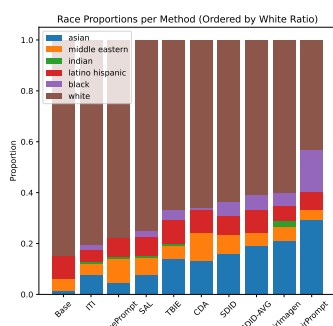 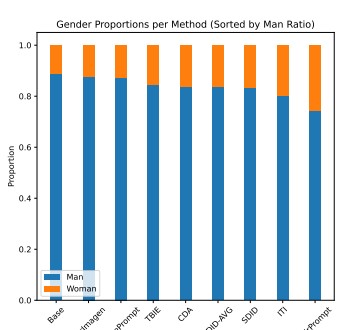

Figure 5: Gender proportions for male-dominated occupations. Each bar shows the proportion of male and female outputs generated by different methods, sorted by male ratio. FairImagen and FairPrompt substantially reduce male overrepresentation compared to baselines.

Figure 6: Race proportions for white-dominated occupations. Each bar shows the proportion of racial groups in the generated outputs, sorted by white ratio. FairImagen and FairPrompt noticeably reduce white overrepresentation and enhance racial diversity.

Figure 7: Gender distribution for prompts with historically male-associated roles. While most methods retain male-dominant outputs, FairPrompt introduces more females, contradicting historical facts. In contrast, FairImagen preserves the intended gender associations.

in the Base model, with FairImagen and FairPrompt increasing representation of Black, Asian, and Latino Hispanic individuals. These results demonstrate that FairImagen effectively mitigates demographic bias in skewed prompts, achieving fairness comparable to FairPrompt while remaining model-agnostic and training-free.

**4.5.6 Robustness to Demographically Determined Prompts.** A major challenge in fairness-aware generative modeling is to ensure that debiasing methods do not compromise semantic fidelity, particularly when prompts inherently reflect justified demographic attributes. In real-world use cases, certain prompts—such as those referencing historical figures or culturally specific roles—are expected to yield outputs with a specific gender association. Overcorrecting in such cases may lead to semantically incongruent or historically inaccurate generations, undermining user trust and model reliability.[3] Therefore, it is essential for fairness interventions to be context-aware and capable of preserving prompt intent when the bias is grounded in legitimate semantics. To this end, we evaluate whether FairImagen can maintain semantic alignment when prompts exhibit strongly determined gender associations. We focus on examples such as "a middle ages blacksmith", "the Pope", and "the King of France", which traditionally imply male representations. Figure 7 shows that across these historically gender-fixed prompts, most models continue to generate predominantly male outputs. Notably, FairPrompt slightly increases the proportion of female representations, even in male-dominant

---

[3] See, e.g., https://www.nytimes.com/2024/02/22/technology/google-gemini-german-uniforms.html and https://www.theverge.com/2024/2/21/24079371/google-ai-gemini-generative-inaccurate-historical.

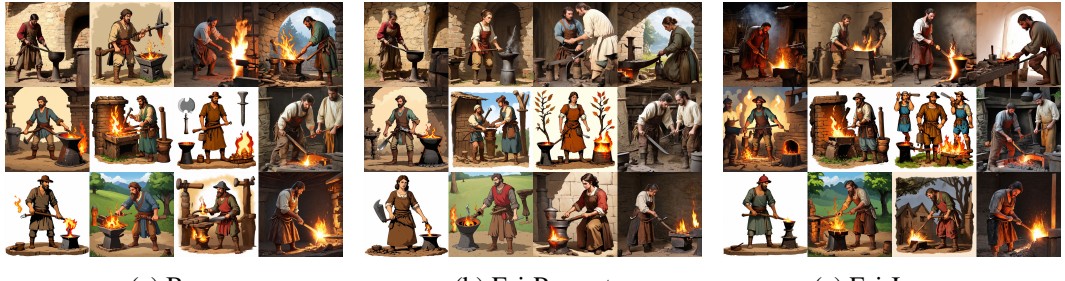

| (a) Base | (b) FairPrompt | (c) FairImagen |

Figure 8: Generated results for the prompt "a middle ages blacksmith" using three different methods: (a) Base, (b) FairPrompt, and (c) FairImagen. FairPrompt introduces female representations even when the prompt implies a male role, while FairImagen preserves the intended gender semantics, yielding historically aligned outputs.

contexts. Figure 8 presents qualitative comparisons of images generated for the blacksmith prompt using the Base, FairPrompt, and FairImagen models. While FairPrompt introduces female depictions regardless of the prompt's semantics, FairImagen respects the gender bias encoded in the original embedding and yields predominantly male outputs. This demonstrates a key strength of FairImagen: when a prompt conveys a strong and contextually justified gender preference, FairImagen does not override it unnecessarily. As such, FairImagen adapts to prompt intent while still being effective in mitigating bias in less explicitly gendered scenarios.

## 5  Conclusion

We present FairImagen, a novel post-hoc debiasing framework for text-to-image generation that integrates FairPCA into the Stable Diffusion pipeline. Our method modifies prompt embeddings to mitigate demographic biases without requiring model retraining or prompt rewriting. Through a fairness-aware projection, empirical noise injection, and a unified cross-demographic formulation, FairImagen achieves strong bias reduction results while preserving visual fidelity and prompt alignment. Extensive experiments across gender, race, and intersectional attributes demonstrate that our approach outperforms existing post-hoc baselines on both fairness and utility metrics. By offering a training-free, model-agnostic, and extensible solution, FairImagen paves the way for more equitable and controllable generative systems.

## Acknowledgements

This work was supported through research funding provided by an EPSRC Doctoral Scholarship, the Wellcome Trust (grant no. 223765/Z/21/Z), Sloan Foundation (grant no. G-2021-16779), Department of Health and Social Care, EPSRC (grant no. EP/Y019393/1), Origin Investments, and Luminate Group.

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

# Appendix. Supplementary Material

## A   Impact Statement

This work contributes to the development of fair and inclusive generative AI systems by introducing FairImagen, a training-free and model-agnostic framework for mitigating demographic bias in text-to-image diffusion models. FairImagen offers a practical and scalable solution for enhancing fairness in image generation without requiring access to model internals or manual prompt design. Its ability to jointly address multiple demographic attributes while preserving visual fidelity makes it particularly well-suited for deployment in real-world applications such as digital media, design, and educational content. By reducing the social harms associated with biased generation and enabling more representative outputs, FairImagen supports the broader goal of responsible and equitable AI deployment.

## B   Ethical Statement

This research aims to address ethical concerns surrounding bias and representation in text-to-image generative models. Our proposed framework, FairImagen, is designed to reduce the amplification of demographic stereotypes without compromising image quality or user intent. We acknowledge that fairness is a multi-faceted and context-dependent concept, and our method focuses primarily on gender and racial representation, which may not capture the full spectrum of social identities or cultural nuances.

We do not collect any personal or sensitive user data in our experiments. All generated images are produced from synthetic prompts, and demographic groupings are based on commonly used protected attributes in fairness research. While FairImagen mitigates certain biases, we caution against interpreting it as a complete solution to fairness in generative models. Ongoing monitoring, inclusive evaluation, and engagement with affected communities remain essential for ensuring responsible deployment.

Our code and findings will be released to the research community to promote transparency and further development of fair and accountable generative AI.

## C   Limitations

Despite the strengths of FairImagen as a post-hoc, training-free debiasing framework, several limitations remain. First, the method currently focuses on a limited set of protected attributes—primarily binary gender and a coarse categorization of race. As frequently noted [48], such simplifications may overlook more nuanced or intersectional demographic identities, such as non-binary gender expressions or multi-ethnic backgrounds. Second, as FairImagen operates on CLIP-based prompt embeddings, it inherits any intrinsic biases present in the CLIP encoder, which itself is trained on large-scale web data with limited curation. While FairPCA reduces group-dependent variance, it cannot fully disentangle bias that is deeply entangled with semantic meaning. Third, although empirical noise injection and projection dimensionality offer tunable fairness-utility trade-offs, determining the optimal balance often requires empirical tuning and may vary across tasks. Additionally, while the framework performs robustly across a wide range of prompts, its effectiveness may degrade for prompts that are strongly tied to cultural or historical contexts, where bias removal risks semantic distortion. Lastly, our evaluation focuses on a specific benchmark of occupational prompts; broader testing across domains, cultures, and creative settings is needed to fully validate generalizability and uncover edge cases where the method may fail.

## D   Effect of Noise Injection Variants

To investigate the role of noise in fairness-aware generation, we compare several noise injection schemes within the FairImagen framework. The **Empirical Noise** method (enoise) samples both direction and magnitude from real group-specific embedding distributions, introducing realistic and data-driven perturbations. **Mean Empirical Noise** uses the average projection magnitude instead

| | Gender Fairness | Accuracy | MUSIQ | Avg |
|---|---|---|---|---|
| Empirical Noise | 0.455 | 0.808 | 0.567 | 0.61 |
| Mean Empirical Noise | 0.432 | 0.788 | 0.472 | 0.564 |
| Fixed Directional Noise | 0.258 | 0.76 | 0.542 | 0.52 |
| Fixed Directional Noise (b=1) | 0.197 | 0.761 | 0.547 | 0.502 |
| Random Gaussian Noise | 0.167 | 0.758 | 0.56 | 0.495 |
| Fixed Random Gaussian Noise | 0.258 | 0.763 | 0.538 | 0.52 |
| Constant Bias Shift | 0.242 | 0.757 | 0.541 | 0.514 |
| Bypass Projection | 0.136 | 0.817 | 0.631 | 0.528 |

Table 3: Comparison of different noise injection strategies used in the FairImagen framework for gender debiasing. Each method perturbs prompt embeddings in distinct ways to obscure demographic signals. The results highlight the trade-off between fairness, semantic accuracy, and visual quality across noise types.

of sampling, resulting in more stable but less diverse shifts. **Fixed Directional Noise** adds binary noise (±1) along the bias direction and optionally biases the sign, simulating controlled reversals in group representation. **Random Gaussian Noise** injects direction-agnostic perturbations, using a Dirichlet-weighted average when multiple groups are involved. Its variant, **Fixed Random Gaussian Noise**, reuses a fixed noise vector for consistency. **Constant Bias Shift** applies a deterministic translation in the bias direction to all embeddings, representing a non-random intervention. Lastly, **Bypass Projection** disables the FairPCA projection and uses the original prompt embedding, serving as an ablation to isolate the effect of projection from noise injection.

The results in Table 3 show that noise type substantially influences the fairness-utility trade-off. Empirical Noise performs best overall, balancing improved fairness with minimal degradation in image quality and semantic alignment. Mean-based perturbations offer more stable behavior but slightly compromise fairness effectiveness. In contrast, direction-agnostic and fixed-noise variants underperform due to their limited alignment with demographic structures. Deterministic shifting introduces consistent but ineffective debiasing, and bypassing the projection leads to high image quality but minimal fairness improvement. These findings underscore the importance of designing context-aware, group-sensitive noise to support effective and reliable debiasing in post-hoc settings.

# E   Qualitative Impact of Empirical Noise

To further understand the role of empirical noise in enhancing fairness without overly neutralizing semantic attributes, we conduct a qualitative case study on firefighter generation. In particular, we compare two male firefighter images produced by FairImagen under different noise configurations. Both images are generated from the same prompt—"Generate a face image of a firefighter"—but differ in whether empirical noise is applied.

As shown in Figure 9, the image generated with empirical noise injection depicts a man with a beard, aligning with natural variations in male appearance. In contrast, the image generated without empirical noise produces a clean-shaven face, which appears overly neutral and lacks realistic diversity. This comparison highlights that without empirical perturbation, FairImagen may suppress group-dependent cues too aggressively, leading to sanitized outputs that obscure important intra-group variation. Empirical noise mitigates this effect by reintroducing sampled group characteristics, such as facial hair, thereby preserving authenticity while maintaining demographic balance. This illustrates the importance of using noise not merely as randomness, but as a mechanism to approximate the true variability within protected groups.

# F   Effect of Empirical Noise Magnitude

To better understand how empirical noise magnitude influences demographic representation, we conduct a controlled experiment in which we vary the strength of empirical noise injection in FairImagen from −5 to 15. In this setting, empirical noise is applied by perturbing the prompt embedding along the learned demographic bias direction, where the scalar magnitude modulates how strongly group-specific features (e.g., gender-related appearance) are emphasized. We generate firefighter images at each noise level and compute the gender proportions in the outputs.

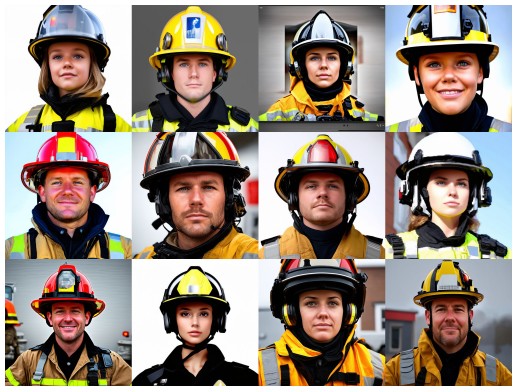 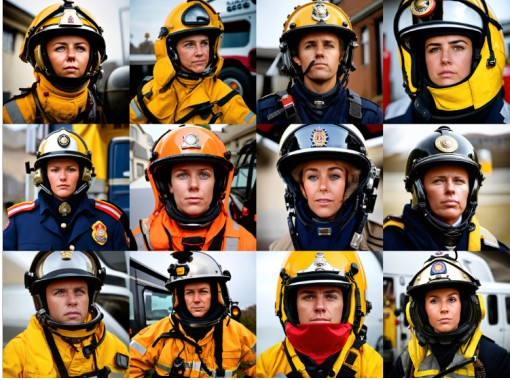

(a) With Empirical Noise           (b) Without Empirical Noise

Figure 9: Visual comparison of firefighter images generated by FairImagen. (a) With empirical noise: preserves natural male features like a beard. (b) Without noise: produces an overly neutral, clean-shaven appearance.

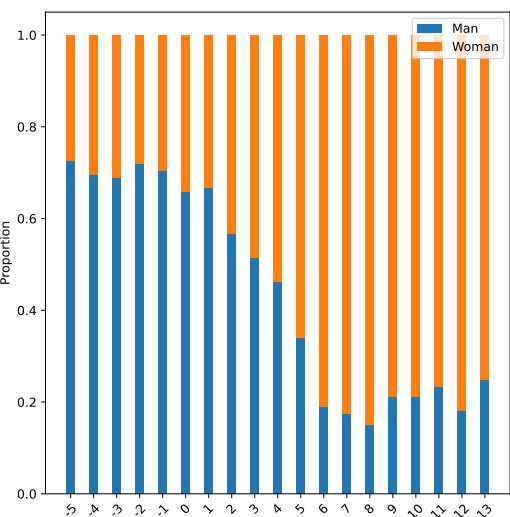

Figure 10: Effect of empirical noise magnitude on gender proportions in firefighter generation. Negative values shift the embedding toward the male-associated direction, while positive values favor female features. The curve is not centered due to male-dominance in the original training prompts.

As visualized in Figure 10, increasing the empirical noise magnitude leads to a clear transition in output gender. Negative values (corresponding to movement in the male-associated direction) produce predominantly male faces, while large positive values favor female representations. Around zero, the output becomes more balanced. However, the inflection point where the transition occurs is not perfectly centered at zero; instead, the gender ratio begins to shift significantly only at slightly positive values. This asymmetry arises because the original training prompts used to construct the FairPCA projection are themselves male-dominated, which results in a skewed embedding space where zero noise still retains residual male characteristics. Consequently, stronger positive perturbations are needed to counteract this bias and achieve gender balance.

This finding reinforces the notion that the empirical bias direction learned by FairImagen captures semantically meaningful demographic information. Moreover, it illustrates how noise magnitude can be used as a controllable parameter to modulate output demographics, with meaningful behavior emerging even from simple linear perturbations.

| | Gender Fairness | Race Fairness | Accuracy | MUSIQ | Avg |
|---|---|---|---|---|---|
| Base | 0.0758 | 0.136 | 0.819 | 0.616 | 0.504 |
| FairPrompt | 0.652 | 0.482 | 0.778 | 0.60 | 0.676 |
| FairImagen (Stack) | 0.227 | 0.345 | 0.799 | 0.569 | 0.532 |
| FairImagen (Sequential) | 0.106 | 0.345 | 0.771 | 0.525 | 0.467 |
| FairImagen (Cross) | 0.429 | 0.325 | 0.792 | 0.601 | 0.596 |

Table 4: Comparison of cross-demographic debiasing methods under joint gender and race settings. FairImagen (Cross) achieves the best fairness-utility trade-off among post-hoc methods.

## G   Comparison of Cross-Demographic Debiasing Strategies

To evaluate the effectiveness of different strategies for handling multiple protected attributes, we compare three cross-demographic debiasing approaches within the FairImagen framework: **Stack**, **Sequential**, and our proposed **Cross** method. The results are presented in Table 4.

The **Stack** method, originally proposed in the FairPCA paper [25], constructs a single fairness projection by stacking group indicator vectors from all demographic attributes (e.g., gender and race) into a combined group matrix. While this approach is simple and computationally efficient, it tends to over-represent groups with stronger bias signals, leading to suboptimal fairness across multiple dimensions.

The **Sequential** method applies FairPCA in multiple stages, projecting out bias directions for each protected attribute one after another. While this offers a conceptually modular way to remove group-specific information, it suffers from compounded information loss: each projection removes components orthogonal to prior ones, making it difficult to preserve semantic fidelity across multiple passes.

In contrast, our **Cross** method explicitly constructs a joint demographic space based on the Cartesian product of all attribute groups (e.g., *White Male*, *Asian Female*), and learns a unified projection that removes shared group-dependent directions in a single step. This ensures that the projection is optimized for intersectional fairness without over-pruning or repeated reconstruction loss.

As shown in Table 4, the Cross method achieves the best balance between gender and race fairness, while also maintaining competitive Accuracy and MUSIQ scores. Compared to Stack and Sequential variants, Cross substantially improves fairness without significantly compromising generation quality, highlighting its effectiveness in multi-attribute debiasing scenarios.

## H   Age Debiasing Experiments

To demonstrate the generalizability of FairImagen beyond binary gender and race categories, we extend our evaluation to age debiasing. Age represents a particularly challenging demographic attribute due to the inherent difficulty in distinguishing precise ages (e.g., differentiating between 35 and 38 years old in generated images). Therefore, we categorize age into three distinct groups: **young** (approximately 18-30 years), **middle-aged** (approximately 31-55 years), and **elderly** (approximately 56+ years).

The age debiasing task follows the same experimental setup as our gender and race experiments, using the same set of occupational prompts and evaluation metrics. We apply our FairPCA-based projection to remove age-correlated variance from prompt embeddings and use empirical noise injection to encourage diverse age representations in generated images.

As shown in Table 5, the results demonstrate that: (1) FairImagen and its variants (FairImagen-T5, FairImagen-OC) achieve the highest fairness scores among all compared methods, confirming the effectiveness of our approach in age debiasing; (2) FairImagen-OC achieves the best overall performance among our proposed variants, indicating strong compatibility with alternative text encoders; (3) all FairImagen variants maintain competitive accuracy and visual quality metrics with only modest degradation compared to the baseline, preserving the generation quality while improving fairness; and (4) the average scores show that FairImagen variants achieve the best overall balance between fairness and utility, validating the effectiveness of our post-hoc debiasing framework across different demographic dimensions beyond binary gender and race categories.

|  | Fairness | Accuracy | MUSIQ | Avg |
|---|---|---|---|---|
| Base | 0.165 | 0.783 | 0.572 | 0.507 |
| FairPrompt | 0.21 | 0.764 | 0.582 | 0.519 |
| ForcePrompt | 0.202 | 0.765 | 0.579 | 0.515 |
| SAL | 0.193 | 0.785 | 0.595 | 0.524 |
| CDA | 0.195 | 0.775 | 0.564 | 0.511 |
| TBIE | 0.403 | 0.748 | 0.557 | 0.569 |
| SDID | 0.253 | 0.779 | 0.545 | 0.526 |
| SDID-AVG | 0.368 | 0.75 | 0.548 | 0.556 |
| ITI | 0.17 | 0.772 | 0.525 | 0.489 |
| FairQueue | 0.318 | 0.749 | 0.617 | 0.562 |
| FairImagen | 0.412 | 0.742 | 0.558 | 0.57 |
| FairImagen-T5 | 0.45 | 0.738 | 0.564 | 0.584 |
| FairImagen-OC | 0.45 | 0.738 | 0.563 | 0.584 |

Table 5: Age debiasing results comparing FairImagen variants against baseline methods. Ages are categorized into young, middle-aged, and elderly groups. FairImagen variants achieve the highest fairness scores while maintaining competitive quality metrics.

|  | Fairness(%) | Accuracy(%) | MUSIQ(%) | Avg(%) |
|---|---|---|---|---|
| Base | 0.61 | 0.05 | 0.39 | 0.24 |
| FairPrompt | 1.85 | 0.14 | 0.46 | 0.66 |
| ForcePrompt | 1.01 | 0.09 | 0.39 | 0.28 |
| SAL | 1.36 | 0.07 | 0.22 | 0.45 |
| CDA | 1.63 | 0.06 | 0.33 | 0.58 |
| TBIE | 1.63 | 0.08 | 0.29 | 0.55 |
| SDID | 2.17 | 0.11 | 0.32 | 0.78 |
| SDID-AVG | 4.3 | 0.5 | 0.46 | 1.71 |
| ITI | 0.76 | 0.11 | 0.36 | 0.34 |
| FairQueue | 1.21 | 0.15 | 0.61 | 0.26 |
| FairImagen | 2.37 | 0.13 | 0.28 | 0.74 |
| FairImagen-T5 | 2.17 | 0.13 | 0.28 | 0.67 |
| FairImagen-OC | 2.27 | 0.13 | 0.28 | 0.72 |

Table 6: Standard deviation values across 10 random seeds for gender debiasing experiments. The results demonstrate the statistical stability and reliability of our experimental findings.

These results validate that FairImagen can be readily extended to additional protected attributes beyond the binary categories typically studied in fairness literature, opening opportunities for more comprehensive bias mitigation in text-to-image generation.

# I   Statistical Stability Analysis

To evaluate the statistical reliability of our experimental results, we conduct additional experiments by varying the random seed from 1 to 10 and computing the standard deviation for each evaluation metric. As shown in Table 6, the results demonstrate that: (1) the standard deviation values are relatively modest across all methods, indicating consistent and reliable experimental results; (2) FairImagen variants exhibit stable performance with standard deviations comparable to other baseline methods, confirming the robustness of our approach; (3) the performance differences between FairImagen and baseline methods are substantially larger than the natural variation introduced by random seed changes, validating the statistical significance of our improvements; and (4) the impact of our debiasing approach on image quality metrics remains within acceptable ranges, demonstrating that our method achieves meaningful fairness improvements while introducing quality changes comparable to the inherent stochasticity of the generation process itself.

