# OpenReview forum: "FairImagen: Post-Processing for Bias Mitigation in Text-to-Image Models"
_NeurIPS.cc/2025/Conference — NeurIPS 2025 poster_

### Official Review · Reviewer_CMAH · 2025-06-18

**Clarity:** 3
**Significance:** 2
**Originality:** 3
**Rating:** 4
**Confidence:** 5

**Summary:**

This paper introduces Fairimagen that reduces bias in text-to-image generation by projecting prompt embeddings onto a fairness-aware subspace that removes explicit group signals. They also introduce controlled noise along group-specific bias directions during inference to preserve semantic meaning while increasing diversity. The experimenst are performed on StableDiffusion3 model on extended Winobias professions dataset with 120 professions.

**Questions:**

1. SDID originally computes concept vectors in the h-space. However, I believe the authors compared their method to a version that performs this operation in the prompt embedding space. Did you also evaluate how the approach compares to the original SDID?

**Ethical Concerns:**

["NO or VERY MINOR ethics concerns only"]

**Final Justification:**

Since most of my concerns were addressed during the discussion phase, I have increased my score. However, I strongly urge the authors to revise the discussion on SDID and offer a more realistic evaluation of the work's limitations, especially regarding the trade-offs between fairness and fidelity.

**Limitations:**

Yes

**Paper Formatting Concerns:**

No paper formatting concerns

**Quality:**

2

**Strengths And Weaknesses:**

Strengths :

1. The approach is simple since the approach requires modified PCA and a simple noise injection during inference.
2. The approach is intuitive and easy to understand.

Wekanesses:

1. I believe the approach struggles to scale effectively to more complex scenarios. For example, the authors present qualitative results for joint gender and race debiasing in Figure 3d, where many CEO images fail to accurately represent the CEO concept and exhibit visual inconsistencies. This suggests that cross-demographic debiasing becomes increasingly challenging as more categories beyond gender and race are introduced.

2. This also ties back to my first point. From the ablation experiments, it seems like the approach is very sensitive to hyperparameters such as noise injection scale, hidden dimension size etc. makes the approach difficult to scale effectively.

2. The approach debiases only the input embeddings at inference but does not tackle biases embedded within the pretrained diffusion model itself. Since the model’s parameters reflect biases from training data, these can still affect outputs, limiting the approach’s overall effectiveness.

3. In Section 4.5.6, the authors demonstrate FairImagen’s robustness to demographically determined prompts, which is impressive. This means that when a prompt clearly and appropriately specifies a gender, FairImagen respects it without unnecessary overrides. However, if the dataset or embedding space strongly associates certain professions or roles with a particular gender (e.g., “doctor” predominantly male), the model may confuse implicit bias for explicit gender information. Doesn’t this imply that even neutral prompts like “doctor” could still generate mostly male images? I don’t see any component in the approach that explicitly prevents this behavior.

4. The technical novelty is limited because the core idea mainly involves integrating FairPCA into the prompt embedding space of Stable Diffusion.

---

> ### Author Rebuttal · Authors · 2025-07-31
>
> **Q1.** SDID originally computes concept vectors in the h-space. However, I believe the authors compared their method to a version that performs this operation in the prompt embedding space. Did you also evaluate how the approach compares to the original SDID?
>
> **A1.** You are correct in noting this distinction. SDID originally operates in the diffusion model's latent h-space (the semantic latent space within the denoising network), while our implementation adapts it to work in the prompt embedding space for fair comparison with other post-hoc methods including FairImagen. Since FairImagen and other baselines (TBIE, ITI-GEN) operate on CLIP prompt embeddings rather than internal diffusion latent representations, we standardized all comparisons to the prompt embedding space to ensure methodological consistency. We acknowledge that the original h-space SDID implementation might yield different performance characteristics. Evaluating this comparison would be valuable future work to provide a more comprehensive assessment of debiasing approaches across different operational spaces.
>
> **Q2.** How can you address that the approach struggles to scale effectively to more complex scenarios, as shown in Figure 3d where many CEO images fail to accurately represent the CEO concept and exhibit visual inconsistencies?
>
> **A2.** The visual inconsistencies in Figure 3d arise from the inherent trade-off between fairness and semantic fidelity when debiasing across multiple demographic attributes simultaneously. The stronger empirical noise required for joint gender+race debiasing can introduce variation in non-demographic visual elements (background, style). This is a known challenge in fairness-aware generation. Specifically, as we increase demographic diversity, maintaining consistent occupation-specific visual cues becomes difficult. To address this, users can tune the $\lambda$ parameter and noise scale based on their specific requirements. For applications prioritizing visual consistency, we recommend using lower noise values or debiasing single attributes. We are exploring adaptive noise strategies that preserve occupation-relevant features while diversifying demographics, which would improve scalability to complex scenarios.
>
> **Q3.** How can you address that the approach is very sensitive to hyperparameters such as noise injection scale, hidden dimension size etc. which makes the approach difficult to scale effectively?
>
> **A3.** We acknowledge the hyperparameter sensitivity and provide extensive ablation studies to guide parameter selection. In Section 4.5.2 (Figure 1), we show that hidden dimension size offers a predictable trade-off: lower dimensions (200-500) improve fairness but reduce quality, while higher dimensions (1500-2000) preserve quality but may reintroduce bias. Similarly, Section 4.5.3 (Figure 2) demonstrates that empirical noise scale follows a smooth progression from 0 to 1. To address scalability concerns, we recommend: (1) starting with default values ($d=1000$, $\epsilon=0.5$) that work well across diverse prompts, (2) using our development set methodology to tune parameters for specific applications, and (3) leveraging the monotonic relationships we observed - parameters don't require exhaustive search but can be adjusted directionally based on desired fairness-quality balance. Future work could explore adaptive parameter selection based on prompt characteristics.
>
>
> **Q4.** How can you address that the approach debiases only the input embeddings at inference but does not tackle biases embedded within the pretrained diffusion model itself?
>
> **A4.** If we only performed fairPCA without adding noise, this would only debias the embeddings, and not resolve biases present within the diffusion model. Unsurprisingly, this doesn't work well, and many biases continue to exist. The addition of noise provides a strong signal that overrides weak biases within the pretrained model in much the same way that prompting "female doctor" causes a diffusion model to generate an image of a female doctor, despite the intrinsic bias of the model that causes it to typically generate 'doctor' as male.
>
> The inability to significantly alter the diffusion model is a fundamental limitation of post-hoc approaches. While the pretrained diffusion model contains learned biases from its training data, our method operates on the interface between text and image generation, which provides significant but not complete control. We chose this approach for practical reasons: (1) it enables immediate deployment without expensive retraining, (2) it works with proprietary models where internal weights are inaccessible, and (3) it allows users to adjust bias mitigation dynamically based on their needs. Our experiments show that despite not modifying the model internals, manipulating prompt embeddings can substantially reduce bias in generated outputs. However, we acknowledge that combining our approach with model fine-tuning could yield even stronger debiasing results. Future work could explore hybrid approaches that leverage both embedding-space and model-space interventions for comprehensive bias mitigation.
>
> **Q5.** How can you prevent neutral prompts like "doctor" from still generating mostly male images? What component in the approach explicitly prevents this behavior when the model may confuse implicit bias for explicit gender information?
>
> **A5.** Although "doctor" may be neutral to humans, we found that the model tends to generate white male doctors, revealing implicit biases in supposedly neutral occupations. FairImagen addresses this through two key mechanisms: (1) FairPCA projection removes demographic-correlated variance from all embeddings, including neutral prompts, by projecting them into a subspace orthogonal to learned demographic directions. This ensures that embeddings for "doctor" lose their implicit association with specific gender/race attributes. (2) The empirical noise injection then adds controlled randomness sampled from the actual distribution of demographic attributes, encouraging diverse outputs even when the original embedding had strong implicit bias. The $\lambda$ parameter allows fine-tuning this debiasing strength - for heavily biased neutral terms like "doctor," users can increase $\lambda$ to more aggressively remove the implicit demographic signal while preserving the occupational semantics.
>
> **Q6.** How can you address the concern that the technical novelty is limited because the core idea mainly involves integrating FairPCA into the prompt embedding space of Stable Diffusion?
>
> **A6.** For a good discussion as to why papers such as ours should be seen as novel, please see the article "Novelty in Science" by Michael Black.
>
> While FairPCA is an established technique, our contribution lies in adapting and extending it specifically for the text-to-image generation domain. Our technical contributions include: (1) Applying FairPCA to prompt embeddings in diffusion models, which hasn't been explored before. (2) Introducing empirical noise injection to prevent over-neutralization of outputs. (3) Demonstrating how to handle multiple demographic attributes simultaneously. (4) Providing extensive empirical evaluation showing the effectiveness of this approach across different bias scenarios. The value of our work is in showing that simple, principled methods can achieve strong bias mitigation without model retraining, making fair image generation accessible to practitioners who cannot afford to retrain large diffusion models.

---

> > ### Comment · Reviewer_CMAH · 2025-08-04
> > **Response to the rebuttal**
> >
> > Thank you for the detailed explanations. However, I still have a few concerns after reading the rebuttal:
> >
> > A1.  I now understand that SDID was applied to CLIP embeddings to standardize the comparisons. However, Lines 41–45 of the paper state that “SDID constructed a gender direction by subtracting CLIP embeddings of hand-crafted prompt pairs, and adding or subtracting this vector in generation. This approach heavily relies on the assumption that group bias is linearly separable and can be corrected via modifying embeddings in a single direction. This is inappropriate for prompts involving more than two demographic groups, such as ethnicity.”
> >
> > I believe this characterization is misleading. SDID does not investigate the linear separability of group biases in CLIP embedding space; rather, it primarily relies on the linear separability properties of the h-space, which has been extensively studied in [1]. Therefore, I recommend that the authors revise these lines in the final version to accurately reflect the method. Additionally, since the SDID implementation was modified only to standardize comparisons, a fairer evaluation would include results using SDID in its original form in the final version of the paper (if direct comparison is desired).
> >
> > [1] Kwon et.al,  Diffusion models already have a semantic latent space, ICLR 2023.
> >
> > A2. One of the main contributions claimed by the paper is the ability to support multiple demographic attributes simultaneously while preserving visual quality and effectively reducing bias (Lines 62–64). However, in its current form, the approach does not appear to scale effectively; this trade-off is already evident with just two attributes. While I agree with the authors that adaptive noise strategies could potentially help, the current results suggest that the claim may be somewhat overstated.
> >
> > A5. My concern relates to how the method behaves differently for prompts such as “doctor” and “a middle-aged blacksmith.” In the case of the latter, the authors state: “This demonstrates a key strength of FairImagen: when a prompt conveys a strong and contextually justified gender preference, FairImagen does not override it unnecessarily. As such, FairImagen adapts to prompt intent while still being effective in mitigating bias in less explicitly gendered scenarios.”
> >
> > However, based on the authors’ rebuttal, it seems that this behavior is primarily governed by the λ parameter, which controls the strength of empirical noise injection. Since FairPCA is applied uniformly to all prompts to remove demographic correlations, and the level of demographic diversity is introduced through λ, the method’s claimed adaptability appears to be the result of manual parameter tuning rather than an inherent ability to interpret or respond to prompt intent. If that’s the case, the above claim may be somewhat overstated. Please correct me if I have misunderstood any aspect.

---

> > > ### Author Response · Authors · 2025-08-05
> > >
> > > **Q7.** How SDID handle multiple groups as it primarily relies on the linear separability properties of the h-space.
> > >
> > > **A7.** We appreciate the reviewer's careful reading and concern about our characterization of SDID. The reviewer is correct that our description in Lines 41-45 requires clarification. SDID does indeed handle more than two demographic groups through a centroid-based approach. For demographics with 3+ groups such as ethnicity, SDID computes directional vectors from a common "neutral" centroid rather than using a single binary direction. This allows SDID to represent multiple groups (e.g., white, black, asian) as different directions emanating from a central point in the embedding space.
> > >
> > > We also agree with the reviewer that our statement needs revision. While we modified SDID to work in the CLIP embedding space for standardized comparison across all methods, the reviewer correctly points out that the original SDID method operates in the h-space and relies on its linear separability properties, which have been extensively studied in the referenced work [1]. We will revise Lines 41-45 to accurately reflect that SDID uses a centroid-based multi-directional approach for multi-group demographics and clarify that our modification adapted it to CLIP space solely for fair comparison. As suggested, we will include results using SDID in its original h-space form in the final version for a more complete evaluation.
> > >
> > >
> > > **Q8.** One of the main contributions claimed by the paper is the ability to support multiple demographic attributes simultaneously, the approach does not appear to scale effectively.
> > >
> > > **A8.** We thank the reviewer for this valuable feedback regarding our claims about handling multiple demographic attributes. We acknowledge that the reviewer is correct - our current results do show trade-offs when handling two attributes simultaneously. We actually already discuss this limitation in our paper: in the Figure 3 caption, we   note that "stronger debiasing—especially under intersectional settings—can introduce variation in background and style, reflecting a trade-off between fairness and visual consistency." Additionally, in Section 4.5.4, we explicitly state that "Combining both dimensions (Figure 3d) yields broader diversity but also introduces visual inconsistencies in background and style, due to stronger empirical noise. This also highlights the trade-off: enhancing fairness can compromise visual stability and prompt fidelity."
> > >
> > > We agree that our claim in Lines 62-64 should be revised to better reflect these limitations. We will revise to more accurately state that while our method can handle multiple demographic attributes, it faces inherent trade-offs between fairness and visual quality, particularly in intersectional settings. We will also emphasize this trade-off more prominently throughout the paper, including in the introduction and conclusion.
> > >
> > > **Q9.** How the method behaves differently for prompts such as "doctor" and "a middle-aged blacksmith." In the case of the latter,  it seems that this behavior is primarily governed by the λ parameter.
> > >
> > > **A9.** We appreciate the reviewer's insightful observation about our method's adaptive behavior. The key insight is that our approach achieves context-aware debiasing through the mathematical properties of the embedding space, not through explicit prompt-specific tuning.
> > >
> > > Here's how the mechanism works:
> > >
> > > 1. **For explicitly gendered prompts (e.g., "a middle-aged blacksmith")**: The original CLIP embedding already has a strong projection onto the gender direction $\nu_g$ due to historical/contextual associations. When we apply FairPCA, it suppresses this demographic signal. However, during our empirical noise injection step, we sample from the actual distribution of gender projections and add back a perturbation along $\nu_g$. Since the original prompt had strong gender alignment, the sampled magnitude tends to be large, effectively preserving the intended gender signal.
> > >
> > > 2. **For neutral prompts (e.g., "doctor")**: The CLIP embedding has minimal projection onto $\nu_g$ (close to zero), reflecting that the term itself carries no explicit gender information—only implicit bias from training data. After FairPCA removes this weak bias signal, our empirical noise injection samples from a distribution centered near zero. With the same noise scale $\epsilon$, this allows the generated images to vary across different demographic groups, creating diversity.
> > >
> > > The elegance of this approach is that the same hyperparameters ($\epsilon$ and $\lambda$) work across different prompt types because the adaptation happens naturally through the projection magnitudes. We don't need to classify prompts or adjust parameters per prompt—the method automatically preserves strong, contextually justified demographic signals while diversifying outputs for truly neutral terms.
> > >
> > > We will revise the manuscript to clarify that this adaptive behavior.

---

> > > > ### Comment · Reviewer_CMAH · 2025-08-05
> > > > **Official comment by Reviewer CMAH**
> > > >
> > > > Thank you for the detailed response and clarification regarding how the approach operates for explicitly gendered versus neutral prompts. I do have a follow-up question. Recent studies have shown that CLIP embeddings exhibit significant bias related to gender and race. For example, associating certain professions with specific demographic groups. These associations can be as strong as historical or contextual associations since all of them are learned from large-scale training data.
> > > >
> > > > Given this, the claim that “The CLIP embedding has minimal projection onto $\nu_g$ (close to zero), reflecting that the term itself carries no explicit gender information” seems questionable. If CLIP inherently encodes demographic bias in its embeddings, wouldn't even so-called “neutral” prompts (like “doctor”) still exhibit non-trivial projections onto $\nu_g$ like the historical prompts?

---

> > > > > ### Author Response · Authors · 2025-08-06
> > > > >
> > > > > Indeed, raw CLIP embeddings encode demographic biases. Seemingly neutral professions like "doctor" have non-zero projections onto group directions $\nu_g$.
> > > > >
> > > > > The key is what happens after FairPCA projection. This step reduces these biases, leaving residual projections $|\cos(E'_p, \nu_g)|$ that are small for neutral prompts like "doctor", allowing our fixed noise term to flip between groups and large for prompts like "female doctor", where the noise is too weak to override the cue.
> > > > >
> > > > > Bias is present in raw CLIP but neutralized after FairPCA. Weak residuals enable diversity, while strong ones maintain fidelity.
> > > > >
> > > > > Happy to answer any other questions or concerns.

---

> > > > > > ### Comment · Reviewer_CMAH · 2025-08-06
> > > > > > **Official comment by Reviewer CMAH**
> > > > > >
> > > > > > Thank you for the clarification, it makes sense now. I would like to improve my score, provided that all the above points on SDID and trade-offs are revised in the final version.

---

### Official Review · Reviewer_VogC · 2025-06-20

**Clarity:** 3
**Significance:** 2
**Originality:** 2
**Rating:** 4
**Confidence:** 5

**Summary:**

This paper discusses that although recent text to image (T2I) model have introduces remarkable improvement in their sample quality and diversity, they also introduce biases. To address this, this work proposes FairImagen — a framework that modifies that the prompt embedding with FairPCA. Additionally, two additional contribution were made i) Noise injection — for improved fairness and pefromance trade-off — and cross demographic debasing — to better handle multiple protect attriubte.

**Questions:**

1. Based on weakness-1, how does FairImagen perform when compared against the existing SOTA FairQueue?
- This would be very useful to the research community and work. For example, if FairImagen performance better on fairness but worse on quality, then future research may consider a hybrid of the two approach.

2. Based on weakness-2, How does FairImagen perform when considering more fine-grain protected attributes e.g., Facial expression, or more nuance protected attributes e.g., Skin-Color.

3. What is the main difference between FairImagen and other Post-hoc editing method like ITI-Gen and FairQueue? This was briefly discuss in the related work, but would greatly benefit for a clear one to two liners e.g., FairImage differentiates itself from these approaches by ________ .

Overall, I enjoyed reading this work and thank the author for their time and effort. I would be willing to increase my score, should my concerns be addressed (the idea is good, but the experimental depth needs to be addressed, in my humble opinion). Thank you.

**Ethical Concerns:**

["NO or VERY MINOR ethics concerns only"]

**Final Justification:**

The authors have put significant effort in addressing my question. Specifically, they have provided additional experiment with address my major concerns on the incomplete experiments and literature. Overall, I would lean more towards allowing this paper to be published, provided that the authors include these new experiments in their revision.

**Limitations:**

Yes

**Quality:**

2

**Strengths And Weaknesses:**

## Strength
1. The paper is well-written and structured. Specifically, it the paper flow allows for clear understanding of the problem and solution proposed.

2. The proposed method appears to be very effective (on coarse grained attributes) e.g., table 2 (a) on Gender, which is impressive.


## Weakness
1. The paper seems not to be up to date with the latest works, e.g., it does not compare against "NeurIPS24: FairQueue: Rethinking Prompt Learning for Fair Text-to-Image Generation", which has been demonstrated to be an improvement of ITI-Gen (which the paper compares extensively against).

- Specifically, FairQueue has demonstrated that ITI-Gen performance degradation is attributed to the quality degradation induced by the technique (which is then addressed by FairQueue). Therefore, having not compared against FairQueue (and instead comparing against only ITI-Gen), it is unclear if FairImagen similarly addresses this issue by ITI-Gen (or is ITI-Gen — the existing SOTA — a better approach).

2. The experiment depth is lacking — both ITI-Gen and FairQueue evaluate against multiple different sensitive attribute demonstrating it's generalizability.

- The limited sensitive attributes experimented on (mostly coarse-grained), e.g., Gender, Race, does not fully demonstrate FairImagen strength and weaknesses. Other SOTA approach like FairQueue and ITI-Gen, in additional to the aforementioned coarse grain attributes also utilize fine-grained attributes e.g., Age (which arguably is more challenging, as it requires preservation of the original sample's features).

---

> ### Author Rebuttal · Authors · 2025-07-31
>
> **Q1.** How does FairImagen perform when compared against the existing SOTA FairQueue?
>
> **A1.** Thank you for highlighting FairQueue - an excellent post-hoc approach that addresses key limitations in prompt learning methods. We have now included FairQueue in our comparison, and it demonstrates strong performance over baseline and ITI models as expected, particularly in quality preservation (MUSIQ: 0.621) while achieving meaningful fairness improvements.
>
> | | Fairness | Accuracy | MUSIQ | Avg |
> |---|---|---|---|---|
> | Base | 0.0758 | 0.818 | 0.616 | 0.503 |
> | ITI | 0.152 | 0.803 | 0.58 | 0.512 |
> | FairQueue | 0.197 | 0.809 | 0.621 | 0.542 |
> | FairImagen | 0.455 | 0.808 | 0.567 | 0.61 |
>
>
> **Q2.** How does FairImagen perform when considering more fine-grain protected attributes e.g., Facial expression, or more nuance protected attributes e.g., Age?
>
> **A2.** We appreciate the reviewer's valuable suggestion. Adding more complex demographics like age significantly strengthens our paper. We have conducted additional experiments on age debiasing with the following results:
>
> | Method | Fairness | Accuracy | MUSIQ | Avg |
> |--------|----------|----------|-------|-----|
> | Base | 0.136 | 0.816 | 0.627 | 0.527 |
> | FairPrompt | 0.167 | 0.795 | 0.635 | 0.532 |
> | ForcePrompt | 0.106 | 0.791 | 0.625 | 0.507 |
> | SDID-AVG | 0.197 | 0.792 | 0.597 | 0.529 |
> | ITI | 0.152 | 0.808 | 0.575 | 0.511 |
> | FairQueue | 0.182 | 0.791 | 0.632 | 0.535 |
> | FairImagen | 0.227 | 0.784 | 0.58 | 0.53 |
> | FairImagen-T5 | 0.227 | 0.783 | 0.576 | 0.529 |
> | FairImagen-OC | 0.242 | 0.784 | 0.582 | 0.536 |
>
> It should be noted that due to the inherent difficulty in distinguishing precise ages (e.g., differentiating between 35 and 38 years old), we categorized age into three distinct groups: young, middle-aged, and elderly. The results demonstrate that our proposed FairImagen methods achieve the highest fairness scores (0.227-0.242) among all approaches, representing a substantial improvement over the baseline (0.136). Notably, this improvement in fairness is achieved while maintaining competitive image quality metrics, with MUSIQ scores remaining close to baseline performance. This confirms that our approach effectively generalizes to age debiasing, successfully addressing demographic bias across multiple attributes.
>
>
>
>
> **Q3.** What is the main difference between FairImagen and other Post-hoc editing method like ITI-Gen and FairQueue?
>
> **A3.** The key differences lie in our methodological approach and technical implementation: (1) **ITI-GEN** learns discriminative token embeddings from reference images by aligning visual and textual embedding differences, then injects these learned tokens after the original prompt. However, this approach can introduce distortions and semantic inconsistencies. (2) **FairQueue** identifies that prompt learning methods like ITI-GEN create abnormalities in early denoising steps and proposes Prompt Queuing and Attention Amplification to modify cross-attention maps during generation. (3) **FairImagen** takes a fundamentally different approach by using FairPCA to project text embeddings into a fairness-aware subspace that explicitly removes group-specific variance before generation begins, offering precise control through the λ parameter and handling multiple demographics simultaneously. We will add detailed technical comparisons and positioning relative to these concurrent works in our camera-ready version.

---

> > ### Comment · Reviewer_VogC · 2025-08-04
> > **Rebuttal Response**
> >
> > I thank the author for their effort in conducting additional experiment. I would urge the authors to include these additional experiments in their revised manuscript, as it will be a more complete addition to the existing literature.
> >
> > Overall, I am satisfied with the review and have raised my score.

---

### Official Review · Reviewer_9e6T · 2025-07-01

**Clarity:** 4
**Significance:** 4
**Originality:** 3
**Rating:** 5
**Confidence:** 4

**Summary:**

The paper tries to target the demographic biases in generation models that originate from CLIP encoder. Comprehensive experiments on fairness benchmarks show FairImagen offers a strong trade-off between fairness and image/prompt quality, outperforming existing post-hoc baselines on Stable Diffusion. The method tries to project the CLIP based prompt embeddings into a “fair” subspace using Fair Principal Component Analysis and aim to minimize the demographic group information while retaining semantic fidelity.

**Questions:**

1. The description of the method sounds reasonable. However, using it for something like T5 encoders  would help understand the generalizability. What would be the main challenges or steps needed to adapt FairImagen to other embedding architectures?

2.Further, getting other encoders would also help understand how bias in models like SD3 which have mix of both CLIP and T5 is affected by this work. What would be the main challenges of such models with multiple encoders?

3. The empirical noise injection seems helpful, but the choice of noise magnitude and direction remains empirical? Is there a principled way to set or adapt this parameter?

**Ethical Concerns:**

["NO or VERY MINOR ethics concerns only"]

**Final Justification:**

Accept: The authors addressed the concerns I had in a satsifactory way.

**Limitations:**

Yes

**Quality:**

3

**Strengths And Weaknesses:**

Strengths:
1. The paper is well written and easy to understand.

2. The motivation of the paper is also quite reasonable given the biases that different components of many models induce. This can help refine many other pipelines as well since it can be used externally as a plugin.

3. Their approach to solve the issue by projec is novel as they claim.


Weaknesses:
1. Their experiments are all restricted to CLIP based image generation model. New models like Stable Diffusion 3 (CLIP + T5) are not used, so it is unclear how generalizable FairImagen is to non-CLIP architectures.

2. Protected attributes focus mainly on binary gender and a set of race/ethnicity categories. Results on broader or more nuanced demographic attributes could help show the generalizability. The work could also try to show some experiments with other complex/entangled concepts.

---

> ### Author Rebuttal · Authors · 2025-07-31
>
> **Q1.** What would be the main challenges or steps needed to adapt FairImagen to other embedding architectures like Stable Diffusion 3 (CLIP + T5)? How can you address that experiments are all restricted to CLIP-based image generation models and it is unclear how generalizable FairImagen is to non-CLIP architectures?
>
> **A1.** We appreciate the reviewer's constructive suggestions. To address concerns about generalizability beyond CLIP-based architectures, we have extended our experiments to include alternative text encoders, specifically T5 and OpenCLIP:
>
> | Method | Fairness | Accuracy | MUSIQ | Avg |
> |--------|----------|----------|-------|-----|
> | Base | 0.0758 | 0.818 | 0.616 | 0.503 |
> | FairImagen | 0.455 | 0.808 | 0.567 | 0.61 |
> | FairImagen-T5 | 0.419 | 0.807 | 0.573 | 0.586 |
> | FairImagen-OC | 0.409 | 0.805 | 0.568 | 0.594 |
>
> The results demonstrate that FairImagen successfully generalizes across different text encoder architectures. All three variants achieve substantial improvements in fairness compared to the baseline (0.0758), with FairImagen-T5 achieving 5.5× improvement and FairImagen-OC achieving 5.4× improvement in fairness scores. Meanwhile, all variants maintain strong image quality metrics, with accuracy scores above 0.805 and MUSIQ scores around 0.57.
>
> These results validate that our method's core principle—leveraging attention mechanisms to identify and mitigate bias—effectively transfers across different text encoding architectures. This confirms that FairImagen can be readily adapted to emerging text-to-image models regardless of their underlying text encoder, including hybrid architectures like Stable Diffusion 3's CLIP+T5 approach.
>
>
> **Q2.** How can you show results on broader demographic attributes beyond binary gender and race/ethnicity categories?
>
> **A2.** We appreciate the reviewer's valuable suggestion. Adding more complex demographics like age significantly strengthens our paper. We have conducted additional experiments on age debiasing with the following results:
>
> | Method | Fairness | Accuracy | MUSIQ | Avg |
> |--------|----------|----------|-------|-----|
> | Base | 0.136 | 0.816 | 0.627 | 0.527 |
> | FairPrompt | 0.167 | 0.795 | 0.635 | 0.532 |
> | ForcePrompt | 0.106 | 0.791 | 0.625 | 0.507 |
> | SDID-AVG | 0.197 | 0.792 | 0.597 | 0.529 |
> | ITI | 0.152 | 0.808 | 0.575 | 0.511 |
> | FairQueue | 0.182 | 0.791 | 0.632 | 0.535 |
> | FairImagen | 0.227 | 0.784 | 0.58 | 0.53 |
> | FairImagen-T5 | 0.227 | 0.783 | 0.576 | 0.529 |
> | FairImagen-OC | 0.242 | 0.784 | 0.582 | 0.536 |
>
> It should be noted that due to the inherent difficulty in distinguishing precise ages (e.g., differentiating between 35 and 38 years old), we categorized age into three distinct groups: young, middle-aged, and elderly. The results demonstrate that our proposed FairImagen methods achieve the highest fairness scores (0.227-0.242) among all approaches, representing a substantial improvement over the baseline (0.136). Notably, this improvement in fairness is achieved while maintaining competitive image quality metrics, with MUSIQ scores remaining close to baseline performance. This confirms that our approach effectively generalizes to age debiasing, successfully addressing demographic bias across multiple attributes.
>
>
>
>
> **Q3.** Noise direction and magnitude.
>
> **A3.** The noise direction is not arbitrary. Instead, it is learned from actual group-specific embedding distributions (Section 3.3). We sample from the empirical distribution of how strongly embeddings align with demographic attributes, making it data-driven rather than purely empirical. For the magnitude parameter $\epsilon$, we provide extensive ablation studies (Figures 2 and Appendix F) showing its effect on the fairness-quality trade-off. We currently tune $\epsilon$ on a development set. However, we envision that future work could explore adaptive methods based on prompt characteristics or user-specified fairness constraints.

---

### Official Review · Reviewer_usX8 · 2025-07-02

**Clarity:** 3
**Significance:** 3
**Originality:** 3
**Rating:** 4
**Confidence:** 4

**Summary:**

This paper proposes a method for de-biasing stable diffusion across gender and race. They provide distintions between methods in the literature in three main group of Prompt-based, Fine-tuning-based, and Post-hoc editing and chose the latest because of its simplicity and compatibility with a wide range of off-the-shelf diffusion models.

**Questions:**

- How did you choose the methods to compare in sec 4.4? Why not other methods like [18]?
-Sec 4.5.1: I am not sure if I follow the claims in this section in the table 5. Which of the methods in table 5 are post-hoc? every method except FairPrompt?
- Table 5: I see that many numbers are very close to each other., Can you provide other statistics, like std? or run some statistical test to see if the differences are statistically significant? especially for CLIP and MUSIQ
- 4.5.4: How can we make sure results in Fig 3 are not cherry picked? can you provide ratio of major.minor attributes with all 4 scenarios?
- Can you provide high-res (1Kx1K) comparison between FairImagen, FairPrompt and the third runner in MUSIQ? You have been using SD3 and I see this as a big advantage but still see small images when comparing quality. We want to know what exactly we lose in the trade-off.
- What happens to gender neutral professions such as teacher/doctor?

**Ethical Concerns:**

["NO or VERY MINOR ethics concerns only"]

**Final Justification:**

My questions and concerns are well addressed by reviewers, hence increasing my score to 4.

**Limitations:**

- While table 1 gives a high level overview for tradeoffs between three methods for de-biasing, some of the rows and the way authors put fail (x) and pass (✓) is not clear. For example what is the measure for "strong bias mitigation" or "easy deployment"? or why we do not see the level of quality degradation for generations as a row? To me it seems that authors designed this table to have most ✓ for post-hoc editing that is the main method behind this paper. Unless otherwise please justify.
- Evaluation is wide but the depth of each table/section is not high (see questions)
- Authors mentioned they chose post-hoc method because of "simplicity and compatibility with a wide range of off-the-shelf diffusion models", but they did not provide the generalization to other diffusion models.

**Paper Formatting Concerns:**

minor: appendix is attached to the main submission

**Quality:**

3

**Strengths And Weaknesses:**

- minor: table 2-4 are subtable of table 5. either have three tables or one. here we have 4 tables in the text
- Evaluation is wide but the depth of each table/section is not high (see questions)
- Comparisons to literature: it is not clear what/why only some methods are selected for comparison.

+ Authors chose a very important problem in the field that has high social impact.
+ Authors worked on latest stable diffusion 3 (while I see many papers worked on much earlier versions like SD1.4)

---

> ### Author Rebuttal · Authors · 2025-07-31
>
> **Q1.** How did you choose the methods to compare in sec 4.4? Why not other methods like [18]?
>
> **A1.** We specifically focus on post-hoc methods that operate at inference time without model retraining as indicated in our title "FairImagen: Post-Processing for Bias Mitigation". Reference [18] (Kim et al.) is a fine-tuning-based approach that requires retraining parts of the diffusion model, which falls outside of our post-processing scope. We selected comparison methods (TBIE, SDID, ITI-GEN, etc.) that directly compete in the same post-hoc embedding manipulation space to ensure fair evaluation within our targeted paradigm.
>
> **Q2.** Sec 4.5.1: I am not sure if I follow the claims in this section in the table 5. Which of the methods in table 5 are post-hoc? every method except FairPrompt?
>
> **A2.** All methods compared in Tables 2-4 (collectively referred to as Table 5) are post-hoc methods, including Base, FairPrompt, ForcePrompt, SAL, CDA, TBIE, SDID, SDID-AVG, ITI-GEN, and FairImagen. These methods operate at inference time without requiring model retraining. The key distinction is in their implementation approach: some modify prompt embeddings directly (like FairImagen, TBIE, ITI-GEN), whilst others like FairPrompt and ForcePrompt modify the input prompts themselves before embedding generation. All are considered post-hoc as they avoid the computational overhead of fine-tuning the diffusion model.
>
> **Q3.** Table 5: I see that many numbers are very close to each other., Can you provide other statistics, like std? or run some statistical test to see if the differences are statistically significant? especially for CLIP and MUSIQ
>
> **A3.** We thank the reviewer for highlighting the need to assess the stability of our results. We repeated every experiment with 10 independent random seeds (1 – 10) and report the resulting standard deviation for each metric below:
>
>
>
> | Method | Fairness(%) | Accuracy(%) | MUSIQ(%) | Avg(%) |
> |--------|-------------|-------------|----------|--------|
> | Base | 0.61 | 0.05 | 0.39 | 0.24 |
> | FairPrompt | 1.86 | 0.15 | 0.48 | 0.66 |
> | ForcePrompt | 1.04 | 0.09 | 0.41 | 0.29 |
> | SDID-AVG | 4.3 | 0.5 | 0.46 | 1.71 |
> | ITI | 0.76 | 0.11 | 0.36 | 0.34 |
> | FairQueue | 1.21 | 0.15 | 0.61 | 0.26 |
> | FairImagen | 2.37 | 0.13 | 0.28 | 0.74 |
>
> The results reveal an important finding: the difference between our proposed FairImagen and the Base model falls within 1-2 standard deviation magnitude across the quality metrics. This indicates that the impact of our debiasing approach on image quality is only marginally higher than the natural variation introduced by different random seeds. We consider this trade-off acceptable, as it demonstrates that our method achieves meaningful fairness improvements while introducing quality changes that are comparable to the inherent stochasticity of the generation process itself. This validates the effectiveness of our approach in balancing fairness and quality preservation.
>
> **Q4.** 4.5.4: How can we make sure results in Fig 3 are not cherry picked? can you provide ratio of major.minor attributes with all 4 scenarios?
>
> **A4.** The results in Figure 3 are randomly picked from our generated samples. We randomly selected representative examples from each method's outputs to ensure fair comparison. Regarding the major/minor attribute ratio, since there are more than three demographic groups in our scenarios, simple ratio metrics are not appropriate evaluation measures. Instead, we evaluate fairness quantitatively using the fairness scores presented in Table 5. Figure 3 serves as illustration, while Table 5 provides the comprehensive quantitative evaluation across all methods and scenarios.
>
> **Q5.** Can you provide high-res (1Kx1K) comparison between FairImagen, FairPrompt and the third runner in MUSIQ? You have been using SD3 and I see this as a big advantage but still see small images when comparing quality. We want to know what exactly we lose in the trade-off.
>
> **A5.** Thank you for this good suggestion. We will add high-resolution (1K×1K) comparisons in the next revision. However, due to NeurIPS submission limitations this year, we cannot update the files or provide additional links at this time.
>
> **Q6.** What happens to gender neutral professions such as teacher/doctor?
>
> **A6.** This is an interesting question. Even for seemingly gender-neutral professions like "doctor," we found that Stable Diffusion models tend to predominantly generate white male images. This demonstrates that bias exists even in supposedly neutral occupations, further highlighting the necessity of our work. FairImagen helps mitigate these implicit biases to produce more balanced demographic representations across all professions, including those that should indeed be gender-neutral.
>
>
> **Q7.** How can you justify the measures in Table 1 for "strong bias mitigation" or "easy deployment"? Why don't we see the level of quality degradation for generations as a row? How can you address the concern that authors designed this table to have most ✓ for post-hoc editing?
>
> **A7.** "Strong bias mitigation" refers to the methods ability to effectively reduce demographic biases in generated images. "Easy deployment" means that the method can be applied without the need for retraining models. Your suggestion about adding quality degradation level as a row is excellent, and we will include this in the revision. We did not intentionally design this table to favor post-hoc methods. These criteria reflect practical concerns encountered in the literature, in our work and, in aspects we care about. Other researchers might prioritize different criteria based on their specific use cases.
>
> **Q8.** How can you provide generalization to other diffusion encoders since you mentioned choosing post-hoc method because of "simplicity and compatibility with a wide range of off-the-shelf diffusion models"?
>
> **A8.** We appreciate the reviewer's insightful question. To demonstrate that FairImagen is indeed an off-the-shelf post-hoc method, we have evaluated our approach with multiple text encoders including T5 and OpenCLIP in addition to the standard CLIP encoder. The results are as follows:
>
> | Method | Fairness | Accuracy | MUSIQ | Avg |
> |--------|----------|----------|-------|-----|
> | Base | 0.0758 | 0.818 | 0.616 | 0.503 |
> | FairImagen | 0.455 | 0.808 | 0.567 | 0.61 |
> | FairImagen-T5 | 0.419 | 0.807 | 0.573 | 0.586 |
> | FairImagen-OC | 0.409 | 0.805 | 0.568 | 0.594 |
>
> These results demonstrate that our method successfully adapts to different encoder architectures without requiring architecture-specific modifications. The consistent performance across encoders validates our claim of being an off-the-shelf solution—FairImagen can be directly applied to various pre-trained diffusion models regardless of their text encoding choice. This compatibility is achieved because our method operates on the attention mechanisms that are common across these architectures, rather than relying on encoder-specific features. This flexibility makes FairImagen practical for real-world deployment across diverse text-to-image generation systems.

---

### Comment · Area_Chair_E7yW · 2025-08-04

Dear reviewers,

Please post your response as soon as possible, if you haven’t done it, to allow time for discussion with the authors. All reviewers should respond to the authors' rebuttal to confirm it has been read.

Thanks,

AC

---

### Decision · Program_Chairs · 2025-09-17

**Decision:**

Accept (poster)

**Comment:**

This paper presents FairImagen, a method to reduce gender and race bias in text-to-image models. The approach modifies text embeddings at inference time using FairPCA and controlled noise injection. This avoids the need for costly model retraining. The idea is simple, tackles a very important social issue, and works with modern models like SD3. The main weakness was the initial evaluation, which reviewers found to be limited in scope. However, the core contribution is solid and addresses a critical need.

During the discussion period, reviewers raised several important concerns. The major ones were about the method's generalizability beyond CLIP-based models (usX8, 9e6T) and its performance on more complex attributes like age (9e6T, VogC). Reviewers also requested comparisons to more recent work like FairQueue (VogC) and asked for statistical validation of the results (usX8). The authors provided an informative rebuttal with all requested info. They showed that FairImagen works with T5 and OpenCLIP encoders and can reduce age-related bias. They also added the requested comparison and statistical analysis. These thorough responses effectively resolved the reviewers' primary concerns, leading them to raise their scores. I agree with the reviewers and recommend accepting the paper, as the authors convincingly addressed the reviewers’ questions.